# Numerical Study on Effects of Flow Channel Length on Solid Oxide Fuel Cell-Integrated System Performances

**Yuhang Liu [1], Jinyi Liu [1,2,*], Lirong Fu [1,2] and Qiao Wang [1]**

[1] Mechanical and Electrical Engineering College, Hainan University, Haikou 570228, China; 21210802000010@hainanu.edu.cn (Y.L.); 993560@hainanu.edu.cn (L.F.); hereisvikki@126.com (Q.W.)

[2] Collaborative Innovation Center of Ecological Civilization, Hainan University, Haikou 570228, China

* Correspondence: 993632@hainanu.edu.cn

**Abstract:** The structural dimensions of the SOFC have an important influence on the solid oxide fuel cell (SOFC)-integrated system performance. The paper focuses on analyzing the effect of the flow channel length on the integrated system. The system model includes a 3-D SOFC model, established using COMSOL 6.1, and a 1-D model of the SOFC-integrated system established, using Aspen Plus V11. This analysis was conducted within an operating voltage range from 0.4 V to 0.9 V and flow channel length range from 6 cm to 18 cm for the SOFC-integrated system model. Performance evaluation indicators for integrated systems are conducted, focusing on three aspects: net electrical power, net electrical efficiency, and thermoelectric efficiency. The purpose of the paper is to explore the optimal flow channel length of SOFC in the integrated system. The results indicate that there is inevitably an optimal length in the integrated system at which both the net electrical power and net electrical efficiency reach their maximum values. When considering the heat recycling in the system, the integrated system with a flow channel length of 16 cm achieves the highest thermoelectric efficiency of 65.68% at 0.7 V. Therefore, there is a flow channel length that allows the system to achieve the highest thermoelectric efficiency. This study provides optimization ideas for the production and manufacturing of SOFCs from the perspective of practical engineering applications.

**Keywords:** SOFC-integrated system; flow channel length; COMSOL; Aspen Plus

## 1. Introduction

In recent years, due to rapid environmental changes, researchers have been searching for clean energy that is more environmentally friendly and sustainable. Solid oxide fuel cell (SOFC)-integrated systems have attracted attention from various countries due to their high power and efficiency advantages [1,2]. By integrating thermal cycle components and SOFCs into a single system, fuel utilization efficiency can be effectively improved, and the issue of residual energy in the waste gas after the SOFCs' reaction can be addressed [3,4]. At the same time, the optimal configuration and operating circumstances of each component in the integrated system have a significant impact on system performance [5]. Currently, the research is mostly focused on increasing fuel cell performance indicators such as power and electrical efficiency by adjusting SOFC structural factors and operating circumstances. In practical applications, the power consumption of system components will also change with the variation in SOFCs' structural factors and operating circumstances. Therefore, achieving the optimal SOFC performance requires considering their operation in actual systems to better meet the practical needs of industrial production.

Through previous research, it is known that structural parameters are one of the important factors affecting SOFCs [6,7]. In order to analyze the influence of structural dimensions on the performance of SOFCs, numerous scholars have developed simulation models of SOFCs in various states [8–10]. Cui et al. developed a two-dimensional model of a flat plate tube segmented tandem solid oxide fuel cell (FTS-SOFC) to analyze the

impact of structural dimensions on the uniformity of cell voltage. It was found that modifying the length of the runner greatly improved the voltage uniformity. The 5 mm FTS-SOFC exhibits the highest maximum power density when other cell dimensions were held constant [11]. Liu et al. constructed various runner structure planar direct ammonia SOFC (DA-SOFC) models to investigate the impact of runner structure on the thermoelectric performance of SOFC bipolar plates. It was shown that the cell performance, such as power density, is optimal for SOFCs with an elliptical channel structure. It was found that increasing the width of the channel cross-section while decreasing the rib width can improve the performance of the cell [12]. An et al. fabricated a segmented-in-series solid oxide fuel cell (SIS-SOFC) with varying cell-to-cell distances to investigate the impact of cell-to-cell distance on voltage loss. The study demonstrated that the ohmic resistance decreased slightly with increasing cell-to-cell distance, while the polarization resistance decreased to a greater extent. The open-circuit voltage maximum power output density and open-circuit voltage were highest when the cells were spaced 2 mm apart [13]. Jee et al. developed several models of SOFCs with varying electrolyte layer thicknesses to assess the influence of electrolyte thickness structural parameters on SOFC single cells. They demonstrated that as the thickness of the electrolyte decreases, the cell's ohmic loss also decreases, leading to an increase in power density. An SOFC stack with an electrolyte thickness of 80 μm demonstrates a high power density. Furthermore, elevated operating temperatures decrease hydrogen consumption [14]. Mushtaq et al. fabricated planar tubular SIS-SOFCs with varying cathode thicknesses to study the impact of cathode thickness on SOFC performance. The study reveals that the optimal thickness of the cathode is 57 mm. Furthermore, it is suggested that a composite cathode with a defined perimeter and geometry may be a potential method to enhance the performance of flat tubular SIS-SOFCs [15]. The above study focuses on the influence of different structural parameters of SOFCs on the output power and efficiency of SOFCs. However, performance analyses from a systemic perspective are lacking.

In practical production, SOFCs are often used as part of an integrated system, rather than being used alone. Many scholars have conducted a parameter analysis of integrated systems for SOFCs [16]. Considering the high construction costs of SOFC-integrated systems, mathematical modeling and simulation methods are generally used [17]. Zhao et al. developed a model of a solid oxide fuel cell–gas turbine (SOFC-GT) system to gain a comprehensive understanding of the optimal integration of the SOFC, gas turbine, and other system components. The model can be enhanced for stability and efficiency by optimizing parameters such as current density, operating temperature, and the fuel utilization factor of the fuel cell [18]. Huang et al. used Simulink and Ebsilon to conduct homeostatic simulations of the SOFC-GT system. The study found the system efficiency increases with operating temperature, while system efficiency may decrease due to the higher fuel utilization [19]. Ameri et al. developed an SOFC-GT model based on the literature from Siemens-Westinghouse company using Aspen Plus. The results of the simulation show that the electricity generation efficiency within the cell tends to increase and then decrease as the fuel utilization rate improves. With minimum fuel input at a 0.85 fuel utilization rate, the efficiency is close to its maximum value [20]. However, they only used a one-dimensional simulation with non-adjustable parameters, which has limitations. In current research on SOFC-integrated systems, most of the analysis focuses on the impact of fuel utilization on the system efficiency and other performance parameters of fuel cell systems. There is relatively little analysis of the impact of SOFCs' structural parameters on the system. At present, SOFC-integrated systems primarily rely on module modeling using commercial software like Aspen Plus, but this approach has certain limitations [21,22]. The system model created by commercial software is challenging to parameterize for a specific component in SOFC, and it is difficult to simulate certain specific functions. Therefore, it needs to be combined with other simulation models that have adjustable parameters to create more realistic system models.

Therefore, this study establishes an integrated system for SOFCs, taking into account the losses of various subsystems, including the SOFCs. The paper specifically focuses on analyzing the impact of the flow channel length of a single cell on the integrated system. A 3D model of SOFCs with different channel lengths was established based on COMSOL 6.1 [23,24]. The aim was to investigate the optimal flow channel length to achieve the best integrated system performance at different cell voltages. This study provides optimization ideas for the production and manufacturing of SOFCs from the perspective of practical engineering applications.

## 2. Description of SOFC-Integrated System

The proposed integrated system proposed mainly comprises an SOFC stack, an air compressor, an afterburner, a fuel compressor, a water pump, a mixer, and three heat exchangers, as shown in Figure 1. The SOFC operating pressure is standard atmospheric pressure. The SOFC system's operational process is as follows.

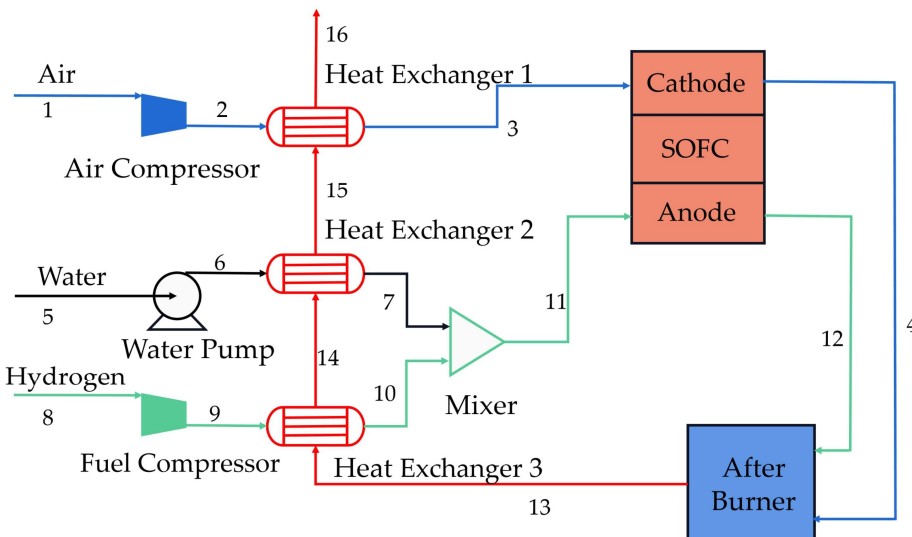

**Figure 1.** Schematic diagram of SOFC-integrated system.

Air temperature is increased to the operating temperature using heat exchanger 1 after being pressurized by the air compressor, and then passed into the SOFC cathode. Water temperature is increased to the operating temperature using heat exchanger 2 after being pressurized by the water pump. Hydrogen temperature is increased to the operating temperature using heat exchanger 3 after being pressurized by the fuel compressor. The mixer mixes heated and pressurized hydrogen and water vapor. The moist hydrogen is sent to the SOFC anode, where it undergoes a chemical reaction with the oxygen, producing an electric current. The unreacted excess gases at the anode and cathode outlets, such as hydrogen, oxygen, and nitrogen, are combined and directed into a combustion chamber for complete combustion, producing a significant amount of heat. The heat is utilized to preheat the hydrogen, water, and air, thus achieving the SOFC operating temperature.

### 2.1. Description of the SOFC Model

Utilizing COMSOL 6.1 software, this paper establishes a 3-D, non-isothermal, numerical model for a single SOFC extracted from the SOFC stack, as depicted in Figure 2a. Air and fuel travel in the flow channel in a counter-current fashion. The model includes equations for momentum, charge, energy, mass, and species conservation, and fully considers the electrode kinetics of electrochemical reactions. To simplify the calculations, the metal interconnects were removed, and the heat transfer around the flow channels was replaced with open periodic heat transfer boundary conditions. At the same time, the anode

electrode was grounded while the cathode electrode was connected to the cell potential. The model geometric dimensions and certain boundary conditions are shown in Figure 2b.

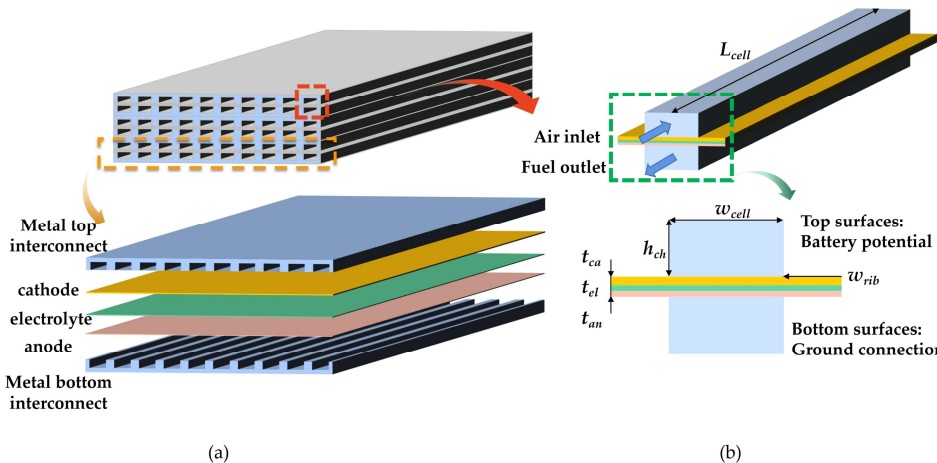

(a)                                                                                     (b)

**Figure 2.** (**a**) Specific structure of SOFC; (**b**) dimensions and boundary conditions. The blue arrow at the top indicates the air inlet, and the blue arrow at the bottom indicates the fuel outlet.

Based on the relevant literature data [14,25], the geometric dimensions of the SOFC are shown in Table 1. Strontium-doped lanthanum manganese oxide/yttrium-stabilized zirconia (LSM-YSZ) is used as the cathode material, and nickel/yttrium-stabilized zirconia (Ni YSZ) is used as the anode material. The electrolyte material unit's mass consists of 92% zirconia ($ZrO_2$) and 8% yttrium-stabilized zirconia (YSZ). This model is established on the COMSOL 6.1 platform, which includes a fuel cell electrochemical model, a porous material transport model, and a heat transfer model. In a bid to better analyze its main issues, this study proposes the following methodology [26,27]:

(1)    The SOFC unit cell operates steadily, and the electrochemical reactions reach equilibrium.
(2)    The cathode and anode are constructed from homogeneous and isotropic porous materials.
(3)    In the anodic electrochemical reaction, only hydrogen gas participates.
(4)    SOFCs maintain consistent flow rates at the cathode and anode inlets over different lengths, and the inlet gas pressure is supplied by a compressor.
(5)    Gas pressure at the anode and cathode outlet is equivalent to the standard atmospheric pressure in the SOFC. Furthermore, the model operates at standard atmospheric pressure.
(6)    The SOFC gas inlet temperature and operating temperature are both set at 800 °C.
(7)    Anode and cathode outlet temperatures are computed from the model.
(8)    All gases are ideal gases in SOFC.

**Table 1.** SOFC model's geometric dimensions.

| Parameter | Value | Unit | Symbol |
|---|---|---|---|
| Channel width | 2 | mm | $w_{ch}$ |
| Rib width | 2 | mm | $w_{rib}$ |
| Anode thickness | 0.15 | mm | $t_{an}$ |
| Electrolyte thickness | 0.1 | mm | $t_{el}$ |
| Cathode thickness | 0.1 | mm | $t_{ca}$ |
| Gas channel height | 2 | mm | $h_{ch}$ |
| Flow channel length | 60–180 | mm | $L_{cell}$ |
| Number of cells | 1000 | - | $N_{cell}$ |

2.1.1. Electrochemical Reaction

Unlike the combustion power generation of internal combustion engines, hydrogen, and oxygen generate electricity directly through electrochemical reactions in SOFC. Oxygen gets electrons from the anode and undergoes a reduction reaction at the cathode, transforming into oxygen ions. The oxygen ions undergo an oxidation reaction with hydrogen after passing through the electrolyte layer, producing electrons and water vapor. This process is accompanied by the movement of electrons and ions, which generates a current. When the reaction reaches equilibrium, the cell current density stabilizes at a constant value, and the voltage can be determined accordingly.

In practice, the actual cell potential is less than the theoretical equilibrium potential due to the effects of various types of cell polarization. The three main polarization losses are activation overpotential, concentration overpotential, and ohmic overpotential. The actual output voltage is shown in Equation (1) [26,28,29].

$$V_{\text{cell}} = E_{\text{r}} - V_{\text{pol}} = E_{\text{r}} - V_{\text{act}} - V_{\text{ohm}} - V_{\text{con}} \tag{1}$$

where $E_r$ is the SOFC reversible voltage, $V_{cell}$ is the SOFC actual voltage, $V_{pol}$ is the total polarization voltage, $V_{act}$ is the activation voltage, $V_{con}$ is the concentration voltage, and $V_{ohm}$ is the ohmic voltage.

In an SOFC, the open circuit voltage (OCV) or the reversible Nernst voltage represent the maximum theoretical voltage that the cell can attain under certain conditions. The reversible voltage of the SOFC is shown in Equations (2) and (3).

$$E_r = E_0 + \frac{R_g \times T}{n_e \times F} \ln\left(\frac{p_{H_2} \times p_{O_2}^{1/2}}{p_{H_2O}}\right) \tag{2}$$

$$E_0 = 1.317 - 2.769 \times 10^{-4} T \tag{3}$$

where $E_0$ is the standard electromotive force at atmospheric pressure, $R_g$ is the gas constant ($R_g$ = 8.314 J/(mol·K)), $T$ is the SOFC operating temperature ($T$ = 800 °C), $F$ is the Faraday constant ($F$ = 96485 C/mol), and $p$ is the fractional pressure for each gas component. $n_e$ is the number of electrons that are transferred ($n_e$ = 2).

Activation polarization is the energy required to overcome the activation energy barrier and allow for an electrochemical reaction to proceed. In the model, activation polarization can be described in relation to the cell current using the Butler–Volmer formula. The Butler–Volmer equation for cathode and anode is shown in Equations (4) and (5) [30,31].

$$i_{an} = i_{0,an}\left[\exp\left(\frac{\alpha_a^a \times n_e \times F \times V_{an,act}}{R_g \times T}\right) - \exp\left(-(1 - \alpha_a^a)\frac{n_e \times F \times V_{an,act}}{R_g \times T}\right)\right] \tag{4}$$

$$i_{ca} = i_{0,ca} \times \left[\exp\left(\frac{\alpha_c^a \times n_e \times F \times V_{ca,act}}{R \times T}\right) - \exp\left(-(1 - \alpha_c^a)\frac{n_e \times F \times V_{ca,act}}{R \times T}\right)\right] \tag{5}$$

where $i_{an}$ and $i_{ca}$ represent the anodic and cathodic current densities. $\alpha_a^a$ and $\alpha_c^a$ represent the anodic transfer coefficient of the anode and cathode. $V_{an,act}$ and $V_{ca,act}$ represent anode and cathode activation polarization. $i_{0,an}$ and $i_{0,ca}$ represent the exchange current densities at the anode and cathode. The cell exchange current density is governed by the mass action law and is expressed as follows:

$$i_{0,an} = \frac{R_g \times T \times K_{an}}{n_e \times F} \exp\left(-\frac{E_{an}}{R_g \times T}\right) \tag{6}$$

$$i_{0,ca} = \frac{R_g \times T \times K_{ca}}{n_e \times F} \exp\left(-\frac{E_{ca}}{R_g \times T}\right) \tag{7}$$

where $E_{an}$ and $E_{ca}$ represent the anodic and cathodic preindex factor. $K_{an}$ and $K_{ca}$ represent the anodic and cathodic activation energy.

Since the SOFC has a high current density in this model, the calculation of the activation loss is simplified by approximation with the Tafel formula [32,33].

$$V_{act} = V_{an,act} + V_{ca,act} = \frac{R_g \times T}{\alpha_a^a \times n_e \times F}(\text{arcsin}h(\frac{i_{an}}{2i_{0,an}})) + \frac{R_g \times T}{\alpha_c^a \times n_e \times F}(\text{arcsin}h(\frac{i_{ca}}{2i_{0,ca}})) \quad (8)$$

The concentration overvoltage can be determined from the following equation [12,34]:

$$V_{con} = V_{an,con} + V_{ca,con} = \frac{R_g \times T}{2F}\ln(\frac{p_{H_2} \times p_{H_2O}^0}{p_{H_2O} \times p_{H_2}^0}) + \frac{R_g \times T}{4F}\ln(\frac{p_{O_2}}{p_{O_2}^0}) \quad (9)$$

where $p_{H_2}$, $p_{O_2}$ and $p_{H_2O}$ represent the fractional pressure of hydrogen, oxygen, and water in working condition. $p_{H_2}^0$, $p_{O_2}^0$ and $p_{H_2O}^0$ represent the fractional pressure of hydrogen, oxygen, and water in the standard state.

During SOFC operation, the flow of electrons in the circuit and the flow of ions in the electrolyte cause ohmic losses, which are calculated as follows [35]:

$$V_{ohm} = iR_{ohm} \quad (10)$$

where $R_{ohm}$ represents the approximate total resistance.

Current density is a crucial parameter for evaluating cell performance. The SOFC model current density is described by Faraday's law [4,36,37].

$$i = \frac{n_e \times F \times \Delta v_{fuel}}{N_{cell} \times A_{cell}} \quad (11)$$

where $N_{cell}$ is the amount of SOFC cells in the stack; $\Delta v_{fuel}$ is electrochemical reaction level (in mol/s); $A_{cell}$ is the active area of a single cell (in m$^2$).

The relationship between electrochemical reaction level and fuel utilization can be expressed by the following equation:

$$\Delta v_{fuel} = v_{fuel} \times U_{fuel} \quad (12)$$

$$U_{fuel} = \frac{i \times N_{cell} \times A_{cell}}{n_e \times F \times v_{fuel}} \quad (13)$$

$U_{fuel}$ is the SOFC fuel utilization rate; $v_{fuel}$ is the hydrogen molar flow rate at the anode (in mol/s).

The single cell output current can be represented as follows:

$$I = i \times A_{cell} \quad (14)$$

The formula for calculating the SOFC stack output power is as follows:

$$W_{SOFC} = I \times V_{cell} \times N_{cell} \quad (15)$$

As shown in Equations (9)–(13), the expression for the SOFC stack output power is as follows:

$$W_{SOFC} = V_{cell} \times U_{fuel} \times v_{fuel} \times n_e \times F \quad (16)$$

To facilitate subsequent data analysis, we introduce the concept of electricity generation efficiency. The efficiency of electricity generation is described as the ratio of the SOFC

stack power output to the fuel energy consumed by the SOFC over a specific period. The expression for electricity generation efficiency is shown in Equation (17).

$$\xi_{ele} = \frac{W_{SOFC}}{\Delta v_{fuel} \times \Delta h_{LHV}} = \frac{V_{cell} \times n_e \times F}{\Delta h_{LHV}} \tag{17}$$

where $\Delta h_{LHV}$ is the hydrogen's lower heating value (241.8 kJ·mol$^{-1}$).

The power generation efficiency of SOFC is represented by Equation (18).

$$\eta_{ele} = \frac{W_{SOFC}}{v_{fuel} \times \Delta h_{LHV}} = \xi_{ele} \times U_{fuel} \tag{18}$$

### 2.1.2. Model Parameter Settings

The SOFC model consists of a hydrogen fuel cell module, a free and porous media flow channel module, a solid and fluid heat transfer module, and a multi-physics field coupling module. Table 2 lists the boundary conditions used in the process of solving the control equations. Table 3 shows the relevant data for the SOFC model in COMSOL 6.1 [38].

**Table 2.** SOFC boundary conditions.

| Parameter | Value | Unit |
|---|---|---|
| Anode electric potential | 0 | V |
| Cathode electric potential | 0.4–0.9 | V |
| Anode air velocity | 0.8 | m/s |
| Cathode air velocity | 3 | m/s |
| Anode mass fraction | $H_2$:$H_2O$ = 0.4:0.6 | - |
| Cathode mass fraction | $O_2$:$N_2$ = 0.15:0.85 | - |
| Anode fuel outlet | 0 | Pa |
| Cathode fuel outlet | 0 | Pa |
| Cell operating temperature | 800 | °C |
| Cell operating pressure | 101.32 | kPa |

**Table 3.** SOFC parameters.

| Parameter | Value | Unit | Symbol | References |
|---|---|---|---|---|
| Anode electronic conductivity | 2149.2 | S/m | $\sigma_{s,a}$ | [39] |
| Cathode electronic conductivity | 5093 | S/m | $\sigma_{s,c}$ | [39] |
| Electrolyte ionic conductivity | 2.2669 | S/m | $\sigma_{ion,l}$ | [39] |
| Anodic transfer coefficient of anode | 0.5 | | $\alpha_a^a$ | [40] |
| Anodic transfer coefficient of cathode | 3.5 | | $\alpha_c^a$ | [40] |
| Anode activation energy | $6.54 \times 10^{11}$ | $1/(\Omega \cdot m^2)$ | $K_{an}$ | [26,41] |
| Cathode activation energy | $2.35 \times 10^{11}$ | $1/(\Omega \cdot m^2)$ | $K_{ca}$ | [26,41] |
| Exchange current density of anode | 4637.4 | A/m$^2$ | $i_{0,an}$ | [26] |
| Exchange current density of cathode | 1166.2 | A/m$^2$ | $i_{0,ca}$ | [26] |
| The specific surface area of anode | 102,500 | 1/m | $S_{an}$ | [42] |
| The specific surface area of cathode | 102,500 | 1/m | $S_{ca}$ | [42] |
| Electrolyte volume fraction | 0.7 | - | $\theta$ | [43] |
| porosity | 0.4 | - | $\varepsilon$ | [26,44] |

It is essential to consider the gas diffusion and the properties of the anode and cathode substrates in the free and porous media flow module. The electrodes' substrate parameters are listed in Table 4. The model considers compressible flow (Ma < 0.3) and Darcy flow but does not account for turbulent flow.

**Table 4.** Substrate parameters.

| Parameter | Value | Unit | References |
|---|---|---|---|
| Porosity of anode | 0.4 | - | [26,44] |
| Porosity of cathode | 0.4 | - | [26,44] |
| Permeability of anode | $1.76 \times 10^{-11}$ | $m^2$ | [26] |
| Permeability of cathode | $1.76 \times 10^{-11}$ | $m^2$ | [26] |

In solid and fluid heat transfer models, it is necessary to define the material properties of the anode and cathode substrates. The material property parameters are depicted in Table 5.

**Table 5.** Material properties.

| Parameter | Value | Unit | References |
|---|---|---|---|
| Anode heat capacity | 450 | J/(kg·k) | [43] |
| Cathode heat capacity | 430 | J/(kg·k) | [43] |
| Electrolyte heat capacity | 470 | J/(kg·k) | [43] |
| Anode density | 3310 | $kg/m^2$ | [45] |
| Cathode density | 3030 | $kg/m^2$ | [45] |
| Electrolyte density | 5160 | $kg/m^2$ | [45] |
| Thermal conductivity of the anode | 11 | W/(m·k) | [43] |
| Thermal conductivity of the cathode | 6 | W/(m·k) | [43] |
| Thermal conductivity of electrolyte | 2.7 | W/(m·k) | [43] |

### 2.1.3. SOFC Model Validation

This paper validates the model using experimental data from the literature. The geometric parameters, material characteristics, fuel characteristics, and SOFC model operating conditions used in this paper are consistent with those in the references [25,26]. Figure 3 compares the simulated current–voltage curve with the experimental current–voltage curve. There is a discrepancy in the results of the comparison, and this discrepancy is due to the fact that there is still a gap between the ideal environment of the model setup and the actual experimental conditions. Generally speaking, the error will be larger in the high-current-density region, which is caused by the temperature difference resulting from ohmic heating during the experiment. This pertains to a general trend and does not impact the accuracy of our results. Therefore, the established SOFC model meets the actual operational needs of SOFCs.

### 2.2. Description of Other Equipment Models in the Integrated System

This paper also describes the properties of other devices in the integrated system. The other equipment mainly comprises heat exchangers, air compressors, fuel compressors, water pumps, mixers, and combustion chambers. To prevent the anode in the SOFC from being oxidized by oxygen, a specific level of hydrogen partial pressure must be maintained at the outlet. Therefore, unreacted hydrogen gas will be present in the anode exhaust gas. The exhaust gases are directed into the combustion chamber to be combusted. The exhaust gas, after combustion, will pass through three heat exchangers consecutively to heat hydrogen, water, and air. This process will fully utilize waste heat, thereby improving the entire system's energy efficiency. To simplify the simulation process, we assume that the various auxiliary modules in the system operate at an isothermal steady state and that the battery system is fully enclosed.

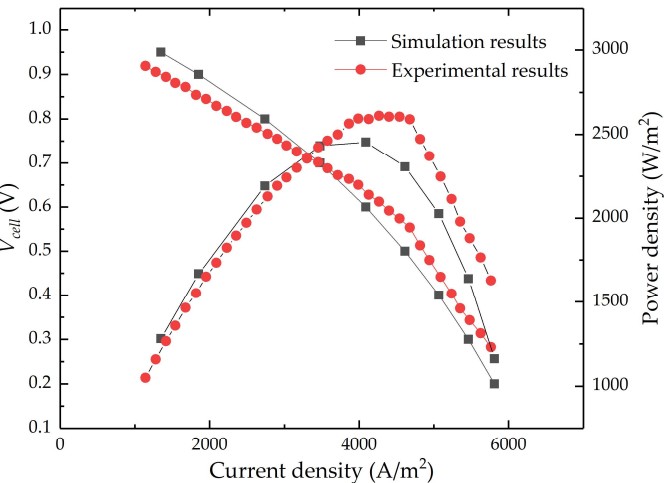

**Figure 3.** Comparison between experimental and simulation results.

The three heat exchangers in the system are used to heat air, water, and hydrogen to the specified temperatures. The heat exchangers used in the system are countercurrent heat exchangers, and the amount of total heat loss stays the same and is not affected by changes in load. The expressions for the heat exchanger model are as follows, using the method described in the literature [45]:

$$\varepsilon = \frac{\left( \dot{M} \times C_c \right) \left( T_{out,c} - T_{in,c} \right)}{\left( \dot{M} \times C_{\min} \right) \left( T_{in,h} - T_{in,c} \right)} \tag{19}$$

where $C_c$ and $C_{min}$ represent the gas-specific heat capacity in the cold stream and the gas-specific heat capacity of the smaller fluid between the hot and cold streams at constant pressure (in J/g·K). $\dot{M}$ is the gas mass flow rate (in g/s). $T_{in,c}$ and $T_{out,c}$ represent the temperatures of the cold inlet stream and outlet stream of the compressor (in °C). $T_{in,h}$ represents the temperature of the hot inlet stream of the compressor. Additionally, when heating water, the latent heat of vaporization for water is considered.

The heat exchange power is strongly dependent on the performance of the equipment and the gas. In an ideal situation, neglecting heat loss, the energy exchange in the heat exchanger is conserved through the change in the thermal energy of the gas. According to this principle, the heat exchanger power can be determined using Equation (20).

$$W_h = U_h \times S_h \times \left( T_{out,c} - T_{in,c} \right) = \left( \dot{M} \times C_c \right) \left( T_{out,c} - T_{in,c} \right) \tag{20}$$

where $S_h$ represents the heat transfer area of the heat exchanger (in m$^2$) and $U_h$ is the convective heat transfer coefficient (in W/(m$^2$·K)).

The fuel compressor and air compressor use the same compressor model. It is assumed that the compressed gas undergoes isentropic compression in an ideal state. Equation (21) represents the formula used to calculate the outlet gas pressure.

$$p_2 = p_1 \times \pi_C \tag{21}$$

where $p_2$ represents the compressor outlet pressure (in Pa) and $p_1$ represents the compressor inlet pressure. $\pi_C$ represents compressor pressure ratio.

According to the definition of isentropic efficiency, the actual outlet temperature can be calculated by Equation (22):

$$\eta_c = \frac{T_2^* - T_1}{T_2 - T_1} \tag{22}$$

where $\eta_c$ is the compressor isentropic efficiency, $T_2^*$ is the ideal outlet temperature of the compressor, $T_1$ is the compressor inlet temperature, and $T_2$ is the compressor outlet temperature. The ideal outlet temperature can be calculated using Equation (23).

$$T_2^* = T_1 \times \pi_C^{\frac{\gamma-1}{\gamma}} \tag{23}$$

where $\gamma$ represents the adiabatic index of the compressor.

The power consumed by the compressor is determined using the following equation:

$$W_c = \dot{M} \times c_p \times (T_2 - T_1) \tag{24}$$

Due to its high density at standard conditions and low flow rate in the pump, water requires relatively low energy to be pressurized to a specified pressure. The pump power consumption is represented by Equation (25).

$$W_p = \frac{M_{h20} \times v_{in}(p_{out} - p_{in})}{\eta} \tag{25}$$

where $M_{h2o}$ is the water mass flow rate (in kg/s); $P_{out}$ and $P_{in}$ refer to the outlet and inlet pressures of the water pump (in Pa); $\eta$ is the water pump efficiency; $v$ is the specific volume of water (in $m^3$/kg).

The combustion chamber is considered to be an ideal combustion chamber, where all the hydrogen gas at the inlet is converted into water. It is also assumed that there is no residual fuel hydrogen at the combustion chamber outlet, and the potential side reactions and oxidative corrosion are not taken into consideration. The combustion chamber is modeled as adiabatic, which means that the enthalpy remains constant at the inlet and outlet, and can be expressed as follows:

$$H_{out}(T_{out}, n_{out}) = H_{in}(T_{in}, n_{in}) \tag{26}$$

Based on the above conditions, an SOFC-integrated system is constructed with a $V_{cell}$ between 0.4 V and 0.9 V and an $L_{cell}$ between 6 cm and 18 cm.

## 3. System Performance Evaluation Indicators

In our effort to accurately assess the system's performance, we established several evaluation metrics, including net electrical power, net electrical efficiency, and thermoelectric efficiency. Among them, the net system electrical power is described as the difference between the SOFC stack's output power and the Balance of Plant (BOP) component's power consumption (auxiliary equipment power consumption), as expressed in Equations (27) and (28).

$$W_{sys} = W_{SOFC} - W_{BOP} \tag{27}$$

$$W_{BOP} = W_p + W_{c,air} + W_{c,fuel} \tag{28}$$

The system's net electrical efficiency is described as the ratio of the system's net electrical output to the low calorific value (LCV) of fuel that enters the system. The representation is shown in Equation (29). This indicator represents the system's output performance.

$$\eta_{sys,net} = \frac{W_{SOFC} - W_{BOP}}{v_{fuel} \times \Delta h_{LHV}} \tag{29}$$

The thermoelectric efficiency of a system is described as the ratio of the system's net electricity and the thermal energy recovered by the system to the LCV of the fuel that enters the system. The recovered heat energy is mainly achieved through the use of a

heat exchanger. The expression is given by Equations (30) and (31). This indicator mainly reflects the efficient use of energy in the system.

$$\eta_{sys,ther} = \frac{W_{SOFC} + W_h - W_{BOP}}{v_{fuel} \times \Delta h_{LHV}} \tag{30}$$

$$W_h = W_{h,1} + W_{h,2} + W_{h,3} \tag{31}$$

## 4. Results and Analysis

This paper is based on the COMSOL 6.1 and ASPEN software. It employs the "3-D numerical simulation of the SOFC stack + 1-D numerical simulation of the system" method to research the effect of different flow channel lengths on the properties of individual components and integrated systems. The paper establishes seven different lengths of SOFC single-cell models, at an $L_{cell}$ ranging from 6 to 18 cm, and analyzes their cell performance at $V_{cell}$ ranging from 0.4 V to 0.9 V. The method of controlling variables was used to conduct analysis and discussion, ensuring the validity of the performance evaluation.

### 4.1. Influence of Different Flow Channel Lengths on SOFC Stack

Figure 4 illustrates the voltage–current and power density profiles of a single cell for various flow channel lengths. Under the same flow channel length, the voltage and current density outputs show an inverse proportional trend, while the power density exhibits a first-increasing-then-decreasing trend with the increase in current density. These findings are consistent with the existing literature [38]. Moreover, as the flow channel length increases, the polarization phenomenon becomes more pronounced, which is consistent with the single-cell performance characteristics. In a single SOFC, the fuel in the anode flow channel is continuously consumed. As a result, the molar flow rate of the fuel gradually decreases, leading to a reduction in the extent of its reaction and a subsequent decrease in the output power per unit area. Therefore, the power density of the SOFC decreases as the size of the runner increases.

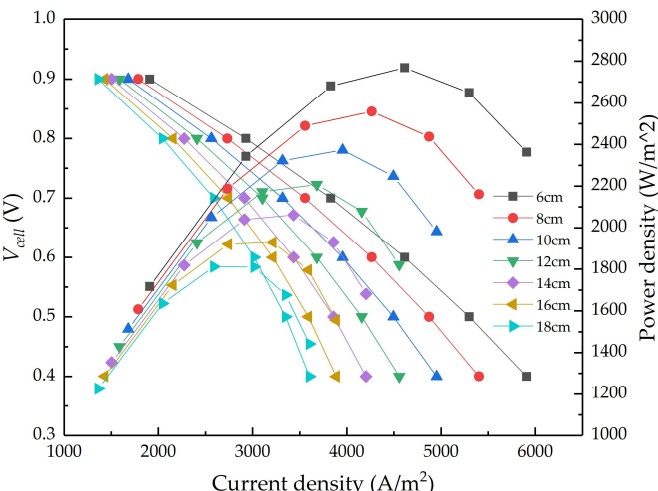

**Figure 4.** Cell voltage–current-power (V-I-P) for all lengths.

Theoretically, increasing the flow channel length will enlarge the reaction area inside the cell. This can improve fuel utilization and subsequently increase the cell output power, assuming a consistent fuel flow rate. The overall power output curve of the SOFC stack is shown in Figure 5. It can be seen from the curve that the output power of the stack increases with the increase in the flow channel length at the same operating voltage. However, the spacing between the curves indicates a gradual decrease in the rate of increase in output power. This decrease is directly related to fuel utilization during the cell reaction process [46].

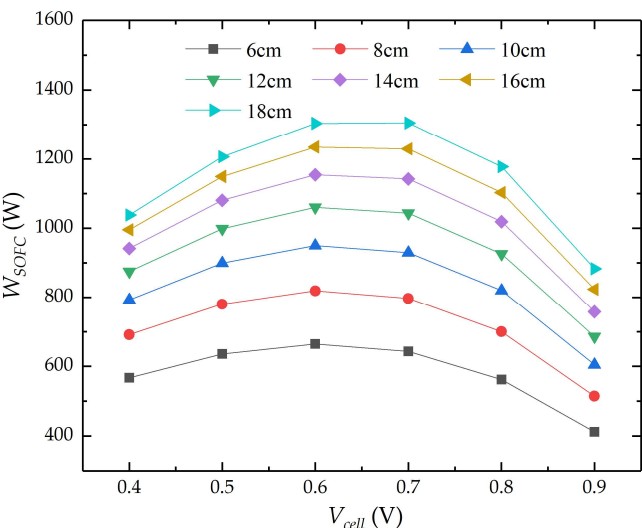

**Figure 5.** Effect on flow channel length and cell voltage on power of the SOFC stack.

Equation (14), which was used for the SOFC output power mentioned above, indicates that when the amount of fuel that is input into the fuel cell remains constant, the SOFC output power is directly proportional to the product of fuel utilization and cell voltage. Therefore, numerical simulations were carried out to analyze the fuel utilization performance at various flow channel lengths, as depicted in Figure 6. The curves indicate that, under the same cell voltage, the fuel utilization rate follows a similar trend to the stack output power as the flow channel length varies. At a working voltage of 0.6 V, the fuel utilization of the SOFC stack increased as the flow channel length increased from 6 cm to 12 cm, rising from 37.45% to 61.87%. This represents a 100% increase in length, while only a 65.2% increase in fuel utilization was recorded. As the flow channel length increases, fuel utilization also increases, but the rate of growth in fuel utilization decreases.

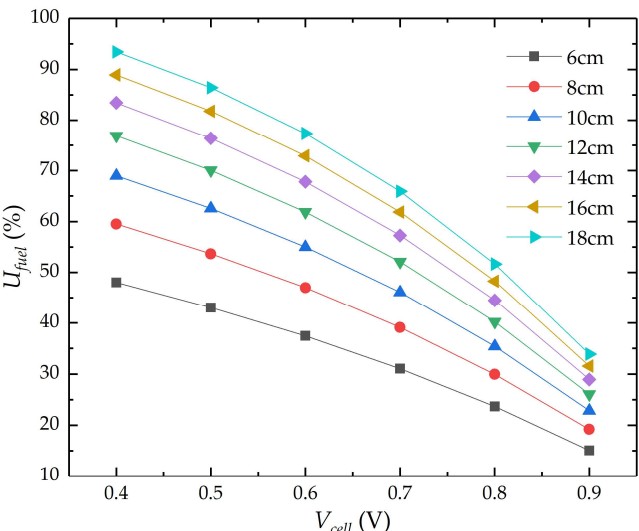

**Figure 6.** Effect on flow channel length and cell voltage on fuel utilization factor.

### 4.2. Influence of Different Flow Channel Lengths on Compressor

The gas pressure in the gas flow path is generated by a fuel compressor and an air compressor, connected to the anode and cathode inlets, respectively. In the SOFC, the flow channel length and the cell operating voltage will affect the gas pressure at the inlets. Through numerical simulation, the air and gas compressors in the integrated system have

pressure ratios and power that vary depending on the flow channel length and operating voltage. This is illustrated in Figures 7 and 8.

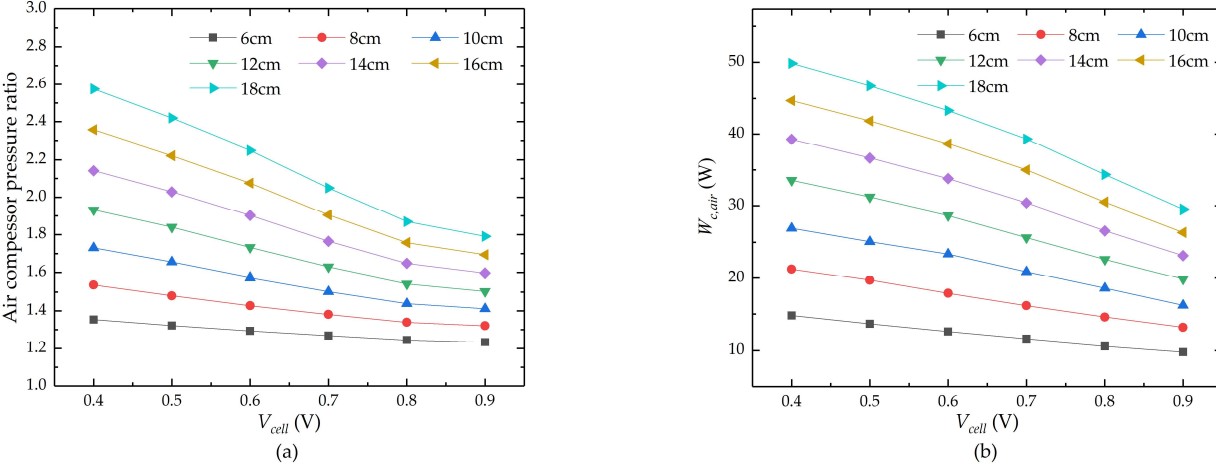

**Figure 7.** (**a**) Effect of flow channel length and cell voltage on the pressure ratio of the air compressor; (**b**) effect of flow channel length and cell voltage on the power of the air compressor.

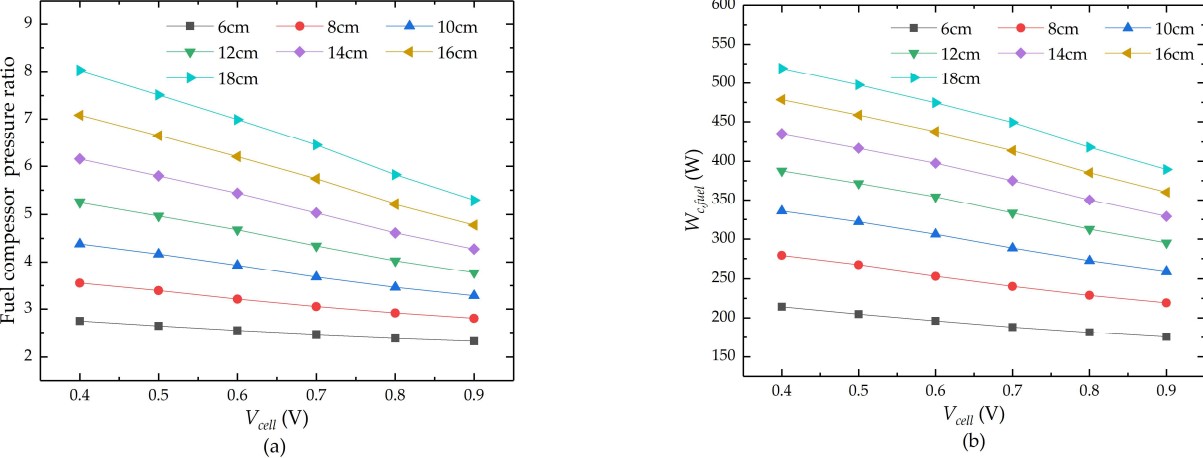

**Figure 8.** (**a**) Effect of flow channel length and cell voltage on the pressure ratio of the fuel compressor; (**b**) effect of flow channel length and cell voltage on the power of the fuel compressor.

It is evident from the curves that both the pressure ratio and power consumed by the fuel and air compressors increase as the flow channel length increases, with the cell voltage remaining constant. To ensure a consistent outlet pressure for fuel cells with varying flow channel lengths, the longer cells require higher air and fuel inlet pressures at the entrance to compensate for the channel-generated pressure drop.

Under the same flow channel length, the ratio of air and fuel compressor pressures and power decrease as the operating voltage increases. However, the decrease in pressure ratio and power is slower with shorter flow channel lengths. This is because, when the SOFC stack operates at a lower cell voltage, it is in a higher current operating condition, leading to higher polarization losses in the cell. From Equation (11), it can be seen that the cell current is directly proportional to the fuel consumption, leading to a higher consumption of hydrogen and oxygen. As a result, there is a greater pressure drop, necessitating higher pressure at the inlets provided by the compressors.

### 4.3. Influence of Different Flow Channel Lengths on Heat Exchangers

The function of heat exchangers in integrated systems is to efficiently harness the residual heat from the exhaust to elevate the temperatures of the fuel and air to the operating level. Heat exchangers 1–3 represent air, water, and hydrogen heat exchangers, respectively. The relationship between power, flow channel lengths, and operating voltages is shown in Figure 9.

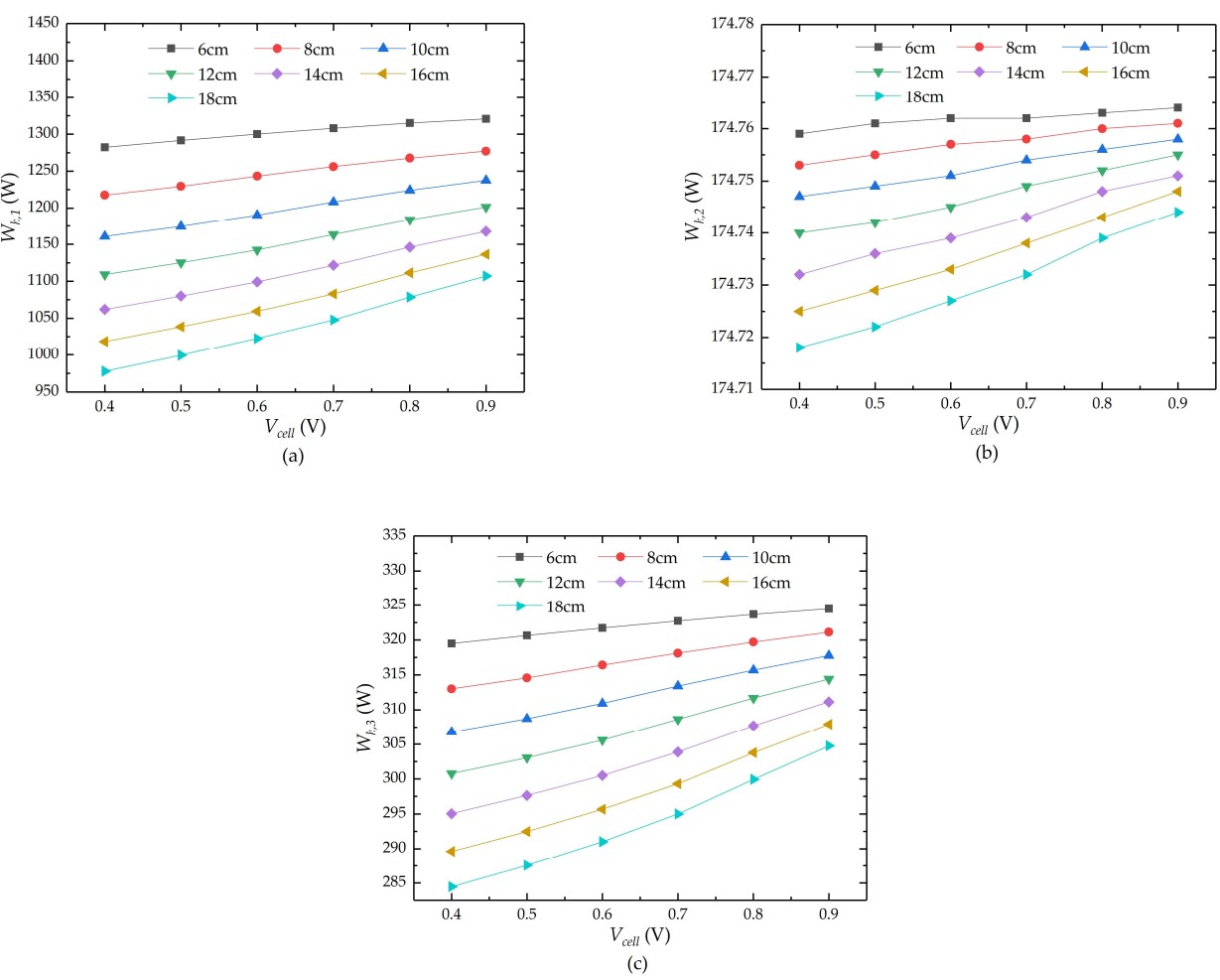

**Figure 9.** (**a**) Effect of flow channel length and cell voltage on the power of the air heat exchanger; (**b**) Effect of flow channel length and cell voltage on the power of the water heat exchanger; (**c**) effect of flow channel length and cell voltage on the power of the hydrogen heat exchanger.

From the curve, it can be observed that there is an inverse relationship between the power consumption of each heat exchanger and the flow channel length at the same operating voltage. The reason for this is that the longer the flow channel length in an integrated system, the more work is required by the compressor to compress the fuel and air. Therefore, the fuel and air attain higher enthalpy values and temperatures at the compressor outlet. The relationship between the outlet temperature, various flow channel lengths, and the operating voltages of the gas and air compressors is illustrated in Figure 10. Therefore, the energy needed to heat the fuel gas and air to 800 °C decreases, resulting in a reduction in power consumption for the heat exchanger. Additionally, Figure 9b shows that the power consumption of the water heat exchanger is minimally influenced by the length and voltage of the cell. This is because when water is pressurized by the water pump at 25 °C, the increase in enthalpy and temperature is minimal. As a result, there is little difference in the energy required for the water to be heated to 800 °C after passing through the heat exchanger.

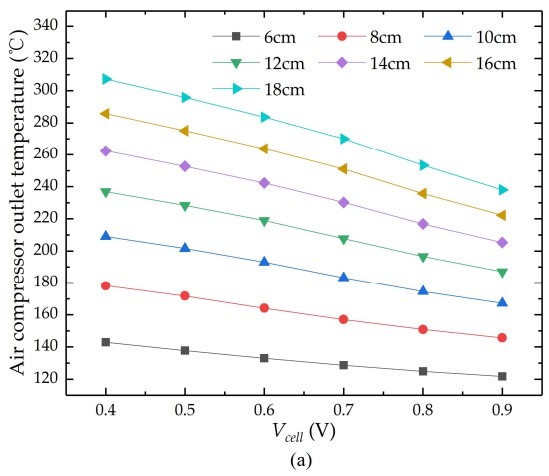
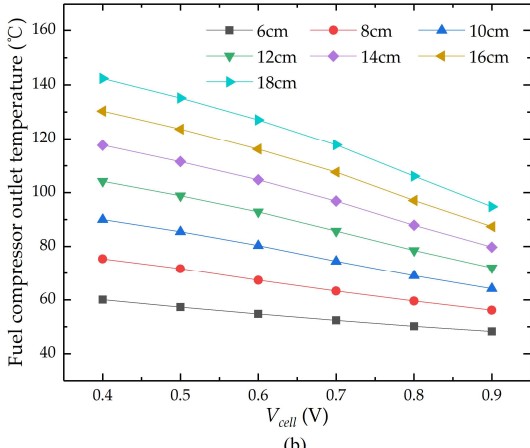

(a)  (b)

**Figure 10.** (**a**) Air compressor outlet gas temperature at different flow channel lengths; (**b**) fuel compressor outlet gas temperature at different flow channel lengths.

### 4.4. System Performances Analysis

From the above analysis, we found that the different flow channel lengths of the SOFC have different impacts on the various components of the integrated system. Therefore, we boldly ask whether there is an optimal length dimension that would result in the integrated system having the best performance. We analyzed the system's optimal performances based on three indicators: net electrical power, net electrical efficiency, and thermoelectric efficiency.

#### 4.4.1. Maximum Net Electrical Power of the System

To minimize the need for external energy inputs, in the integrated system, the SOFC stack supplies electrical energy to other equipment components. The system output power is derived as the difference between the SOFC output power and the power consumed by other energy-consuming equipment in the BOP. The relationship curve between BOP power consumption and different flow channel lengths and operating voltages is shown in Figure 11a, and the relationship curve between the system net electrical power and different flow channel lengths and voltages is shown in Figure 11b. In Figure 11b, when the operating voltage is 0.7 V, the length of the cell ranges from 6 cm to 18 cm, and the system net electrical power shows a gradually increasing trend, but at a slower rate. As the length increases, the SOFC output power also increases. The power consumed by the air compressor also increases simultaneously. However, the growth rate of stack output power is slower than the growth rate of blowout preventer consumption power, leading to a slower growth rate of net system power.

According to Figure 11, we can further determine the correlation between the flow channel length and the system's net electrical power, as shown in Figure 12. At 0.4 V, the overall system power exhibits a rising and then falling trend. The maximum net electrical power is 471.84 W when the length is 16 cm, with a fuel utilization rate of 88.93%. This means that a longer flow channel length does not necessarily result in a better performance. Furthermore, it can be inferred that there is an optimal length value that maximizes the system's net electrical power at various voltages.

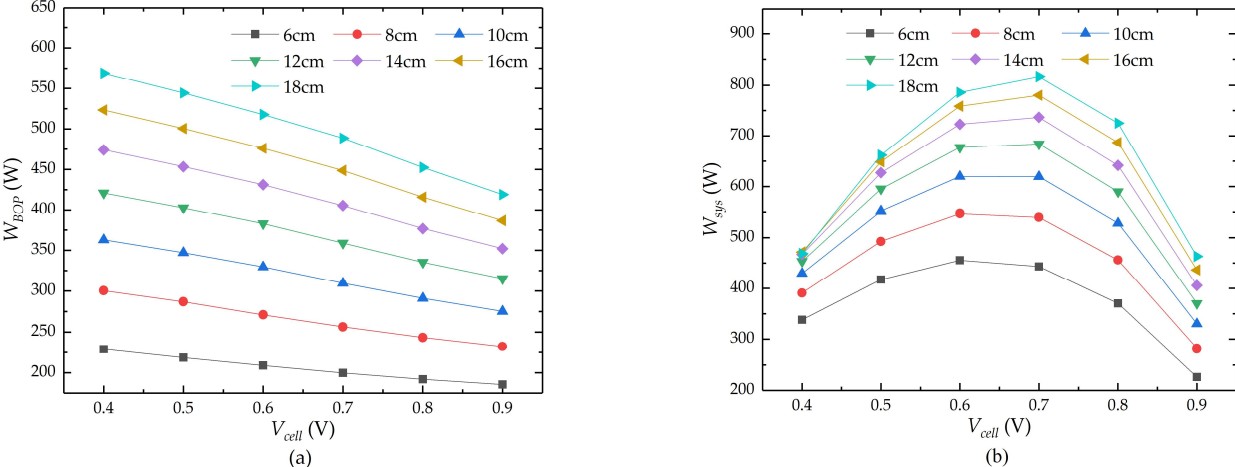

**Figure 11.** (**a**) Effect of flow channel length and cell voltage on BOP power consumption; (**b**) effect of flow channel length and cell voltage on net electrical power.

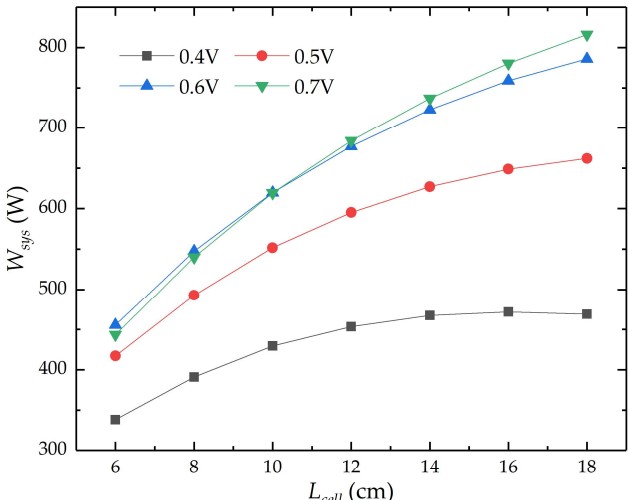

**Figure 12.** Net electrical power.

4.4.2. Maximum Net Electrical Efficiency of the System

On the basis of the previous expressions for the system performance evaluation metrics, the highest electrical efficiency corresponding to the maximum system output power that can be achieved by the SOFC-integrated system is noted when the energy input to the system is kept constant. Therefore, the tendency of the system's net electrical efficiency mirrors the tendency of the system's net electrical power, as depicted in Figure 13. As the flow channel length increases, the system's net electrical efficiency demonstrates a rising but decelerating trend.

Similarly, the changing trend of the system's net electrical efficiency with flow channel length is further illustrated in Figure 14. At 0.4 V, the net electrical efficiency shows an increasing trend with the increase in flow channel length, followed by a decrease. At a length of 16 cm, the maximum net electrical efficiency is 13.25%. It can be inferred that there is an optimal length value for different voltages that maximizes the net electrical efficiency of the system.

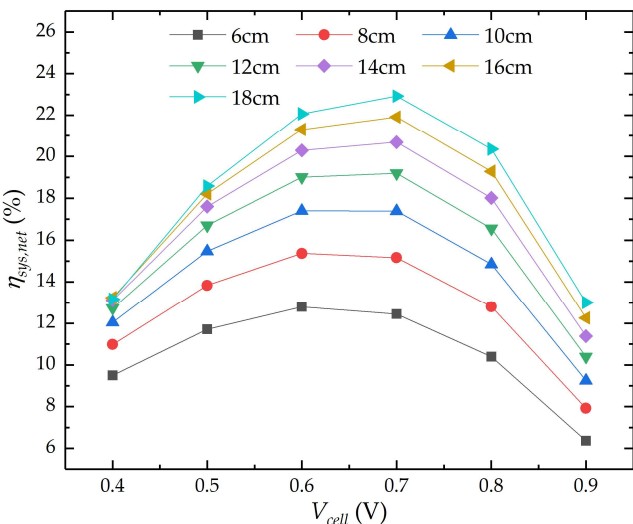

**Figure 13.** Effect of flow channel length and cell voltage on net electrical efficiency.

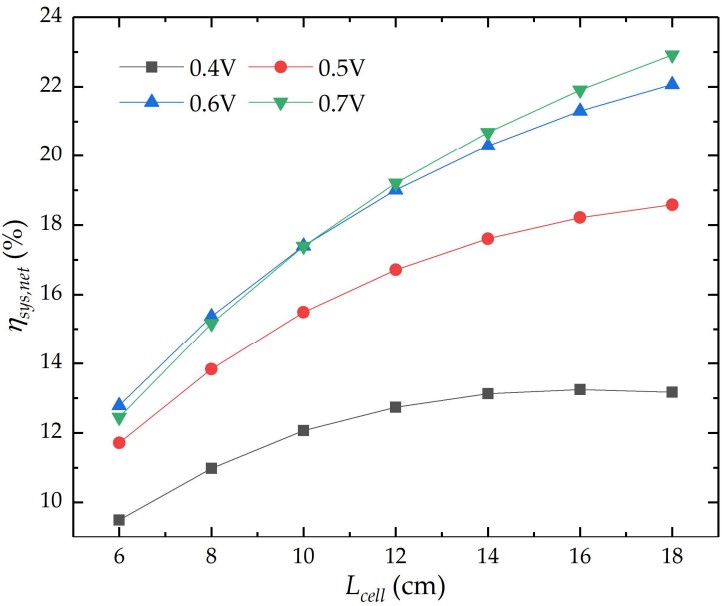

**Figure 14.** Net electrical efficiency.

### 4.4.3. Maximum Thermoelectric Efficiency of the System

From the above discussion, it is evident that, when considering only the power aspect, the system's electrical efficiency is relatively low. However, when considering the utilization of waste heat from the fuel exhaust gases, the integrated system energy efficiency is improved. The thermoelectric efficiency of the integrated system, obtained through numerical simulation, with the variation in flow channel length, is depicted in Figure 15.

In Figure 15, it can be observed that 0.6 V and 0.7 V are close to the optimal operating voltage. With the increase in flow channel length, the system thermoelectric efficiency tends to increase and then decrease. However, the optimal length of a single battery corresponding to the highest thermal efficiency varies at different battery voltages. At 0.6 V, the thermoelectric efficiency of the integrated system is highest when the length is 12 cm, reaching 64.62%. At 0.7 V, the system thermoelectric efficiency is highest when the length is 16 cm, reaching 65.68%. Additionally, the fuel utilization rate is 65.93%, which aligns with the practical engineering situation of SOFCs. At this point, the system's net electrical power output is 816.19 W. This is because, as the flow channel length increases,

the fuel utilization rate of the SOFC also increases, thereby increasing the SOFC output power. Although the energy in the SOFC exhaust gas decreases at this point, there is still a significant amount of energy available to meet the requirements for fuel and air preheating. Longer stacks require higher air and fuel pressure at the inlet, which inevitably increases the compressor power consumption. As a result, the heat transfer power of the heat exchanger is reduced, leading to a decrease in the energy-utilization efficiency of the exhaust gas. Therefore, there is an optimal length at different voltages that maximizes the system's thermoelectric efficiency.

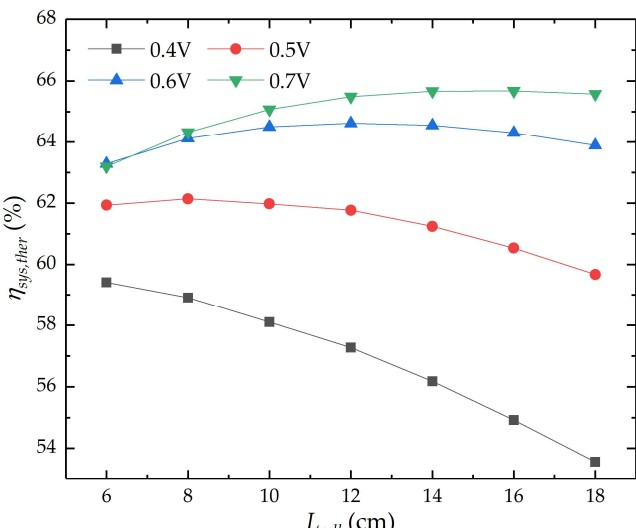

**Figure 15.** Effect of flow channel length and cell voltage on thermoelectric efficiency.

## 5. Summary

This paper investigates an integrated system based on SOFCs. An SOFC was simulated using the multiphysics simulation software COMSOL 6.1 and verified by experimental results. The paper analyzed the impact of the length of a single SOFC cell on system performance, and the results are presented as follows.

Under the same cell voltage, the single flow channel length is directly proportional to the pressure ratio and power consumption of the air and fuel compressors. However, it is inversely correlated with the heat transfer power of each heat exchanger. For an SOFC stack, the variation in single flow channel length affects the SOFC output power and efficiency. Increasing the flow channel length can improve fuel utilization to some extent, thereby enhancing the SOFC output power. For the integrated system, it is inferred that there is an optimal length that maximizes the net electric power and net electric efficiency. When considering heat recycling in the system, an integrated system with a single flow channel length of 16 cm achieves the highest thermoelectric efficiency of 65.68% at 0.7 V. Therefore, the flow channel length is an important factor affecting the performance of the integrated system in real industrial production. This study will provide a theoretical foundation for the further optimization of SOFC power generation systems. In the following study, the proposed numerical model will be combined with a more realistic production-integrated system to design a superior-performing SOFC-integrated system.

**Author Contributions:** Methodology, software, and writing—original draft preparation, Y.L.; conceptualization, funding acquisition, and supervision, J.L.; investigation, writing—review and editing, L.F.; validation, formal analysis, Q.W. All authors have read and agreed to the published version of the manuscript.

**Funding:** This research was funded by the Hainan Provincial Natural Science Foundation of China: 520RC540 and 521RC492; and the Collaborative Innovation Center Project of Hainan University: XTCX2022STC16.

**Institutional Review Board Statement:** Not applicable.

**Informed Consent Statement:** Not applicable.

**Data Availability Statement:** The data presented in this study are available on request from the corresponding author.

**Conflicts of Interest:** The authors declare no conflict of interest.

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
