# Peer review of "Numerical Study on Effects of Flow Channel Length on Solid Oxide Fuel Cell-Integrated System Performances"

_sustainability, doi:10.3390/su16041643_

Round 1

Reviewer 1 Report

Comments and Suggestions for Authors

This paper focuses on analyzing the effect of the flow channel length on a Solid oxide fuel cell (SOFC) integrated system using COMSOL and a 1-D model of the SOFC integrated system established using Aspen Plus.

Some substantial technical changes are required to fully understand the mathematical model introduced in section 2:

       First of all, we recommend a diagram with boxes and arrows to describe the inputs and outputs of the modules of the proposed mathematical model. This is extremely important if you want to understand, for example, what are the input data, what are the intermediate variables and what are the outputs (for example, this would clarify if the variables introduced on the left side of the equations (5) and (6) are intermediate variables or not. Are they output variables?).

       Many of the variables used in the formulations are not described later on, nor their units. See for example “n” in (11) or “deltah_LHV” in (15) or “T_out,c” in (17) or “p_1” in (19). Solve this problem on all sides of the section.

       On line 162 a Greek notation is used which is not consistent with the one used in equation (4). Solve this problem on all sides of the section.

       Always include, as far as possible, the mathematical nomenclature of the variables in the paragraphs when referring to them (as it will be also suggested in the study cases section), so that the reader can relate concepts to mathematical variables (for example, to understand what variable is the Flow channel length, see line 295).

       The variable in the left hand side of (16) appears to refer to two apparently distinct concepts (see lines 202 and 204).

       Set the coherence between the variables used in the equations and in the descriptive paragraphs, because sometimes the mathematical nomenclature does not coincide (for example on line 174 the subindexes of E and K do not appear, which do appear in (7) and (8)). Solve this problem on all sides of the section.

       On lines 178 and 179, what does it mean that V_pol is a parameter that varies from 0.1V to 0.6 V?, What implications does this have with respect to the fact that V_pol is the sum of the variables V_act, V_ohm and Vcon? What implications does it have on (4)?

       In equations (27) and (28) two different mathematical expressions are used to refer to the same variable. Need a clarification.

       Indicate the values of all the constants used in the section (for example, for T and F on line 156). Solve this problem on all sides of the section.

Likewise, with respect to the study cases, use the mathematical notation implemented in section 2 (as much as possible) in the illustrations and graphics, in such a way that the reader can easily connect the results analyzed in the study cases, with the mathematical formula.

Minor changes:

·         COMSOL is not described in line 74. Please introduce at least a reference to understand that COMSOL is a software.

·         Use the same sign of multiplication (it is recommended to use the dot). Also, include in the formulation the sign of multiplication where it is necessary.

·         When making an enumeration of variables, use commas instead of full points (as you did on line 162).

·         Superindex on line 332.

·         Introduce the meaning of the abbreviations used in the abstract and in the conclusion section the first time they appear (for instance SOFC in the abstract, what does it stand for?).

·         Missing spaces between words and brackets, like in “performance[3]” in line 33.

·         E_r, is it the theoretical or the reversible voltage? (see lines 147 and 152).

·         In different parts of the paper, the character “:” is used as expecting a set of bullet points or steps in the following lines (see for example line 83 or line 492). This is not common when writing.

Reviewer 2 Report

Comments and Suggestions for Authors

In this manuscript, the authors summarize the effect of channel length of SOFC on the system by provide comprehensive experimental data. Major revision is recommended:

1.     Some background descriptions are very rough thus the background section needs to be modified. For example, what is the kind of performance that is affected by electrolyte thickness and the operating temperature on page 2 line 52? What are the major limitations that Aspen Plus has as it is the primary module modeling for SOFC on page 2 line 70.

2.     The logical transfer from the background to the issue that this paper tried to address is confusing. In the background section, the author should summary the current issue that they focus on instead of listing the previous work. And the authors should clearly point out their strategy to address the issue.

3.     In Figure 3, what is the reason that the experimental voltage becomes higher than simulated voltage at higher current density? Please also provide the simulated and experimental results for I-P curve (current density and powder density).

4.     In Figure 4, the colors for 6 cm are different for V-I and I-P curves. And in the analysis of the data, the authors emphasize too much on the general trends. The ratio of PPD improvement by tuning the flow channel lengths should be more important. Pls explain why power density increased with short flow channel.

5.     The author claimed that higher cell output power could be achieved via longer flow channel, which should be used to explain the result in Figure 5 instead of Figure 4.

6.     The trend is over extended for the conclusion that therefore, we can infer that there must be an optimal length to achieve the highest fuel utilization and, consequently, the highest output of the SOFC stack for Figure 6. The fuel utilization ratio trends to be higher with longer channel length. But there is no inflection point in Figure 6. So nobody knows if the “zero” length is required for the highest fuel utilization ratio.

7.     Please explain why SOFC stacks consume more hydrogen and oxygen when operated at lower battery voltages for Figure 7 and 8.

8.     Please explain why the peak net electrical powers are obtained at the same voltage (0.7 V) for SOFC with all channel length in Figure 11? According to Figure 4, different peak powder densities are observed at different current density with different channel length. The same question can be found for Figure 13.

9.     List the components of SOFC system along with the operation procedure on page 2 line 80. In Figure 1, remove the arrow within the lines, for example, in line 4.

10.  In Table 2, apply subscript formatting to H2. In Table 3, apply superscript formatting to m2.

Round 2

Reviewer 2 Report

Comments and Suggestions for Authors

The authors have made substantial improvements to the revised manuscript and have provided thorough explanations for the comments raised. Based on these enhancements, I recommend accepting this revised version for publication in Sustainability.